# FedSycle: Mitigating Post-Unlearning Performance Inconsistency in Federated Learning via Latent Feature Decoupling

## Abstract

Federated Learning (FL) safeguards data privacy by enabling collaborative model training without centralizing client data. The emerging 'Right to Be Forgotten' mandates necessitate Federated Unlearning (FU), allowing clients to revoke their data's influence on the global model. However, a critical yet overlooked challenge in FU is the emergence of performance inconsistency across clients following an unlearning event. When a client departs, the global model's accuracy can degrade unevenly for the remaining participants, leading to unfairness and disincentivizing collaboration. To address this, we propose FedSycle, a novel FU framework that leverages the power of pre-trained models to do fast retraining and enhance performance consistency. FedSycle operates by decoupling client data into distinct latent representations: one capturing semantic content (retained locally for privacy and to boost client-side retraining efficiency) and another capturing domain-specific attributes (e.g., texture, color). Crucially, only the less sensitive domain attributes are aggregated on the server. The server then utilizes these aggregated attributes to synthesize auxiliary data, which guides the global model update, effectively recalibrating its performance across all remaining client domains. We provide theoretical convergence guarantees for FedSycle. Extensive experiments on standard benchmarks (PACS, DomainNet) demonstrate its superiority. FedSycle not only achieves state-of-the-art unlearning effectiveness but also significantly mitigates performance inconsistency, reducing its variance by up to 83.2% compared to leading baselines, while simultaneously improving the average accuracy for non-target clients by over 31%.

## 1 Introduction

Federated Learning (FL) is a distributed machine learning paradigm (McMahan et al., 2017) designed to preserve participating clients' data privacy and security. In a typical FL training round, the server distributes the global model to clients, who perform local training using their private data. The server then collects model parameter or gradient updates from these clients and aggregates them to update the global model. This approach enables collaborative learning while obviating raw data exchange, thus enhancing the model's performance and generalization capabilities.

However, sole focus on training-phase data privacy in FL systems remains insufficient. Due to the fact that the FL global model would implicitly remember clients' local data, it's necessary to implement the 'Right to Be Forgotten' (RTBF) regulation (Chik, 2013; GDPR, 2016; Mullman, 2017), an ideal FL system should empower clients to control their data contributions post-training. This requirement has spurred Federated Unlearning (FU), which develops mechanisms for models to "unlearn" target data, thereby exhibiting behavior indistinguishable from never having encountered it. To this end, an effective FU algorithm must satisfy two critical objectives (Liu et al., 2024b): (1) complete removal of target data influence, and (2) maintenance of model utility through performance recovery.

Existing FU algorithms, including those leveraging historical information (Liu et al., 2021; Wu et al., 2022a; Zhang et al., 2023), reverse learning (Wu et al., 2022b; Zhao et al., 2023b), and techniques such as clustering and model merging (Gu et al., 2024b; Su & Li, 2023), have made significant strides in balancing these objectives. Nevertheless, they encounter substantial challenges in

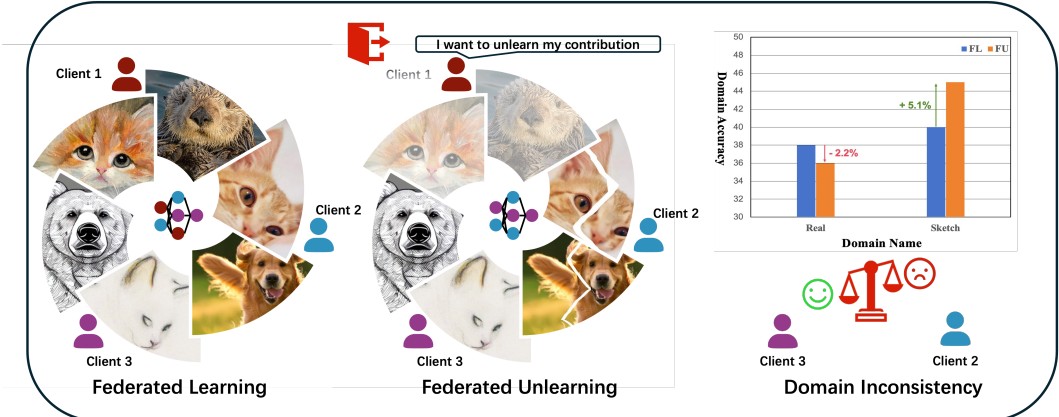

Figure 1: Illustration of Federated Unlearning and the corresponding Domain Inconsistency. Three clients jointly train a FL model. When client 1 (due to its own will, contract, etc., decides to withdraw), the FU process is initiated and erases client 1's data contribution. However, because the data of client 1 is coupled to varying degrees with other clients, the performance of the unlearned model varies across different domains, resulting in experiences differences and causing potential unfairness.

scenarios characterized by domain heterogeneity—a common feature of real-world non-independent and identically distributed (Non-IID) settings. Domain heterogeneity arises when different clients possess data from different domains, even for the same label. As shown in Fig.1, a single class label (e.g., "cat") may span heterogeneous domains (sketches, real images, paintings). This domain-specific data distribution is central to the unlearning challenge. When a client exits and invokes FU, the global model exhibits asymmetric performance shifts across non-target clients' domains—a phenomenon we term Domain Inconsistency. Quantitatively, unlearning reduces real-image domain accuracy by 2.2% but paradoxically improves sketch-domain accuracy by 5.1% (Fig.1, right). This bidirectional divergence creates fairness concerns, as clients experience domain-dependent performance disparities, violating the uniform service guarantee expected in FL systems. Our theoretical analysis (Appendix B) traces this bias to entangled feature representations in high-dimensional space, where unlearning target client unintentionally perturbs non-target clients' decision boundaries.

To address these challenges, we draw inspiration from recent advances in FL that leverage pre-trained models (PTMs) to mitigate data heterogeneity while preserving privacy. Building on this, we propose integrating PTMs into FU to alleviate Domain Inconsistency. However, to the best of our knowledge, there exists no work that integrates PTMs with FU, and existing methods combining FL and PTMs still incur additional training overhead—such as GAN-based approaches—and raise privacy concerns, particularly when uploading complete local data representations. These limitations motivate us to develop a more secure and efficient approach that harnesses the power of PTMs while upholding the privacy guarantees of FL.

To this end, we first introduce Domain Inconsistency, a mathematically rigorous metric to quantify cross-domain performance degradation (Def. 1). Leveraging this metric, we design FedSycle, a novel dual-side FL framework with style-content decoupling. FedSycle enables clients to retain sensitive content features and to do feature fusion locally, significantly improving retraining efficiency. On the server side, the framework leverages clustered style information (e.g., color, texture) to restore domain-specific performance. This approach effectively reduces Domain Inconsistency while maintaining strict privacy preservation.

In this work, we make the following key contributions:

- We are the first to discover and mathematically define Domain Inconsistency—a novel phenomenon in FU where unlearn a client's data causes inconsistent performance changes across heterogeneous domains.
- To mitigate Domain Inconsistency, we propose FedSycle, a dual-side FU framework with style-content decoupling. On the client side, sensitive content features are preserved locally to enhance classification performance, while the server uses updated style features to synthesis auxiliary data to reduce Domain Inconsistency. FedSycle not only reduces Domain Inconsistency to ensure fairness but also guarantees privacy, with theoretical convergence guarantees.

- Our experiments demonstrate that FedSycle reduces Domain Inconsistency by 83% compared to baselines with better non-target clients' accuracy and unlearn efficiency.

## 2 RELATED WORKS

### 2.1 FEDERATED UNLEARNING

Federated unlearning (FU) is a critical mechanism designed to selectively erase the global model's knowledge of target client data, while maintaining the model's utility (e.g., accuracy or client-specific objectives). Beyond fulfilling the 'RTBF' mandate in federated systems, FU provides an adversarial defense capability by neutralizing poisoned data contributions—effectively acting as both a privacy safeguard and security enhancement for FL ecosystems.

**FU Targets.** Existing works categorize FU targets into three levels (Liu et al., 2024b; Zhao et al., 2023a): sample-level, class-level, and client-level. These levels require FU algorithms to unlearn a specific subset of data, a particular class, or all data from certain clients. Recently, more finegrained local feature unlearning has also been achieved (Gu et al., 2024a). This paper focuses exclusively on client-level unlearning, where the goal is to remove all traces of a participating client's data contribution from the global model. While sample-level and class-level unlearning are important directions, client-level unlearning addresses critical scenarios in FL, such as client revocation (e.g., due to privacy regulations or contractual expiration) or adversarial participation. We argue that client-level unlearning is a foundational challenge with unique requirements, distinct from fine-grained unlearning settings.

**FU Algorithms.** Due to the strict privacy constraints in FL, no part of the system can expose clients' raw data (Li et al., 2020; Zhang et al., 2021), making traditional machine unlearning methods difficult to apply in FU scenarios. Although directly retraining with the remaining clients is the golden approach, the process incurs substantial time and computational costs, significantly diminishing clients' motivation. Therefore, FU methods primarily focus on approximate unlearning, which can generally be categorized into the following types: (1) **FU based on historical information**. These works have leveraged the historical information of FL training to erase the target data contribution, such as (Liu et al., 2021; Wu et al., 2022a; Zhang et al., 2023). (2) **FU based on reverse learning**. These methods implement unlearning by simulating the reverse process of learning, such as gradient ascent (Li et al., 2023a; Wu et al., 2022b). (3) **FU based on others**. Previous works have also explored other approaches, such as model merging (Gu et al., 2024b) and clustering (Su & Li, 2023).

Although many advanced algorithms exist, the Domain Inconsistency brought by unlearning remains overlooked. The testing domain performance of non-targeted clients diverges from that prior to unlearning. More critically, the extent of the change is highly likely to differ across clients, directly leading to disparities in experience and creating potential inequities, which in turn undermines the enthusiasm for continued participation in training. Therefore, we aim to design a novel FU algorithm which can significantly recover domain performance degradation, and reduce Domain Inconsistency thereby enhancing the utility and robustness of the unlearned model.

### 2.2 MITIGATING DATA HETEROGENEITY THROUGH PTMs IN FL

Data heterogeneity presents a fundamental challenge in Non-IID FL scenarios (Rahman et al., 2021), where skewed data distributions across clients lead to suboptimal global model convergence and impaired generalization performance (Li et al., 2020). These limitations ultimately reduce client participation incentives (Pan et al., 2024; 2023; Wen et al., 2023), threatening the sustainability of FL ecosystems. In this context, pre-trained models (PTMs) offer a compelling solution through their dual capabilities: (a) cross-task transferable representation learning and (b) heterogeneity-mitigating data augmentation. This synergy makes PTMs naturally suitable for FL integration, simultaneously leveraging their superior performance characteristics while preserving the privacy and efficiency requirements. Existing PTM-enhanced FL approaches fall into three principal categories:

**Parameter-based PTMs.** FeDGEN (Zhu et al., 2021) extracts client-specific information from classification layers to train a GAN model, generating latent features enriched with global information for data augmentation. FedCGAN (Xiao et al., 2024) trains client-specific generator parameters using

local discriminators, then aggregates them into a global generator that synthesizes and distributes supplementary data to participants.

**Request-based PTMs.** WGAN-GP (He et al., 2025) requires clients to contribute local real samples for GAN training, followed by on-demand data generation tailored to client clusters' requirements, while FIMI (Li et al., 2023c) necessitates clients to upload partial data features for GAN training, subsequently performing conditional data generation based on individual clients' supplemental data requests.

**Data Feature-based PTMs.** SlaugFL (Liu et al., 2024a) filters local class prototypes, uploads core prototypes to the server, and generates supplementary data via GANs. FedDISC (Yang et al., 2024) updates local class prototypes to interact with the server's ground-truth information for label generation, proposing a novel federated semi-supervised learning paradigm with diffusion model.

While existing approaches demonstrate promise, they suffer from three critical limitations: (1) the computational overhead from auxiliary model training (e.g., GANs) significantly increases system costs; (2) the requirement for inter-client information sharing creates both privacy leakage risks and practical coordination barriers; and (3) the extraction and transmission of class prototypes inevitably exposes sensitive client data, including labels and discriminative image features. These fundamental constraints make current methods unsuitable for seamless integration with unlearning algorithms while effectively addressing Domain Inconsistency. Our method further decomposes the feature information of the local image into style and content features, the clients retain the privacy-sensitive content features (subject, labels), and only uploads the style features (color, texture, etc.) that cannot be reversed to restore the original image to boost unlearning process, maintain model utility, and reduce Domain Inconsistency simultaneously.

## 3 PRELIMINARIES AND METRICS

In this section, we introduce the FL setting considered in this work and the corresponding basic FL paradigm in Sec. 3.1. The process and objectives of FU are stated in Sec. 3.2. And we abstract the definition of Domain Inconsistency into the mathematical expression in Sec. 3.3.

### 3.1 BASIC FEDERATED LEARNING

We consider the horizontal FL scenario, and the typical optimization paradigm can be formulated as follows:

$$\min_w \left\{ F(w) \triangleq \sum_{i=1}^N p_i F_i(w) \right\}, \tag{1}$$

where $N$ is the total number of clients, $F_i(w) \triangleq \mathbb{E}_{\xi \sim D_{\text{train}}^i} [F_i(w, \xi)]$ is the local objective with training data distribution $D_{\text{train}}^i$ of client $i$. $w$ is the model parameters, and $p_i$ represents the aggregation weight for client $i$, satisfying $\sum_{i=1}^N p_i = 1$.

### 3.2 FU PROCESS AND OBJECTIVES

The federated unlearning algorithm consists of two main stages:

- **Target Data Unlearning**. Given a federated model $w^o$ well-trained on the dataset $\mathcal{D}$ (as addressing unlearning requests prior to well-trained model is of limited meaning) and a dataset $D_u \subset D$ that belongs to the target clients and needs to be unlearned. The FU algorithm must first erase the data knowledge from global model regarding $D_u$ at this stage.
- **Performance Recovery**. Executing target data unlearning may affect the model's generalization performance, so the FL system will perform additional performance recovery operations. Typically, this involves conducting several additional training rounds with the non-target clients according to the following objectives:

$$w^u = \arg\min_w [F_{\mathcal{D}/\mathcal{D}_u}(w)], \tag{2}$$

where $F_{\mathcal{D}/\mathcal{D}_u}$ is the global objective function within the non-target clients.

### 3.3 DOMAIN INCONSISTENCY

Generally, the evaluation of FU can be assessed from three dimensions: unlearning efficiency, unlearning effectiveness, and model utility after unlearning. However, the metrics above are insufficient

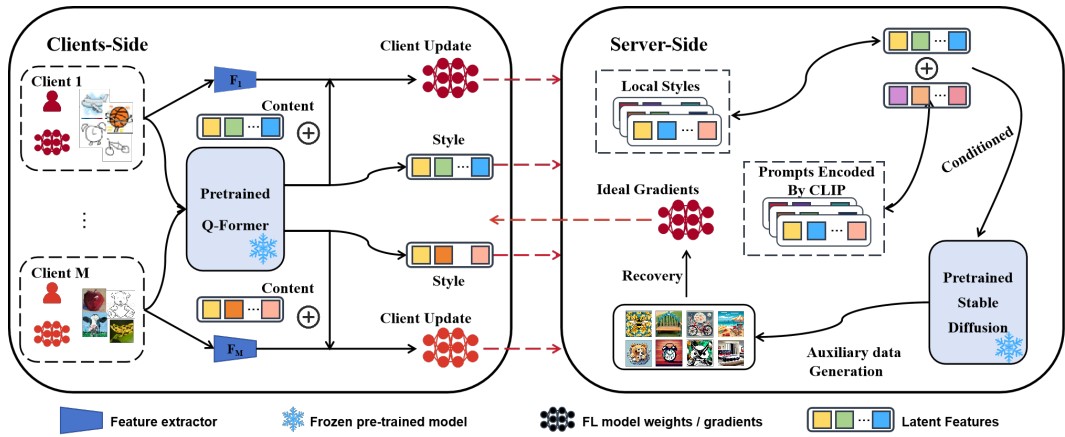

Figure 2: The framework of FedSycle. The overall method consists of four steps: (a) decoupling of image content and style; (b) local classification enhancement via content features; (c) server-side auxiliary data generation; (d) performance recovery training.

to represent, assess, and improve the potential Domain Inconsistency challenges that may occur during the FU process. To bridge this gap, we provide an mathematical expression to represent the Domain Inconsistency phenomenon as follows.

**Definition 1 (Domain Inconsistency, $DI$)** *Let $\Phi$ denote the set of test data domains of all non-target clients, $\mathcal{D}_{test}^u$ denotes the test data of target clients, $D_r = \{x \mid x \in \mathcal{D}_{test}/\mathcal{D}_{test}^u\}$, $D_\phi = \{x \mid x \in D_{test}/\mathcal{D}_{test}^u, Domain(x) = \phi\}$, and $Var(\cdot)$ denotes the variance calculation. For a given well-trained FL model $w^o$ and the global model $w^u$ obtained after unlearning, the Domain Inconsistency is defined as follows:*

$$DI(\boldsymbol{w}^u, \boldsymbol{w}^o) = Var(\{|\frac{F(D_\phi; \boldsymbol{w^u}) - F(D_\phi; \boldsymbol{w^o})}{F(D_r; \boldsymbol{w^u}) - F(D_r; \boldsymbol{w^o})}|\}_{\phi \in \Phi}). \tag{3}$$

We combine the trends of individual domain changes with the average trends of overall domain changes. The variance value reflects the inconsistency of the relative change trends of different domains. Therefore, the smaller the DI, the closer the relative changes of domains are, indicating a smaller inconsistency, and vice versa.

**Remark 1** *This DI metric effectively assesses the phenomenon of Domain Inconsistency, incentivizing unlearning algorithms to pursue better model utility after unlearning (as reflected in the denominator), while minimizing variations in domain performance (as indicated in the numerator). A lower DI value indicates that the impact of unlearn behavior on the overall performance of global model is consistent, suggesting that no additional domain preferences or discrimination are introduced due to unlearning. This represents higher system fairness, client satisfaction and more robust model generalization performance.*

## 4 METHODOLOGY

FedSycle is a fast, retraining-based FU algorithm that addresses Domain Inconsistency while enhancing model utility. As depicted in Fig. 2, FedSycle utilizes a pre-trained text-image Q-former model (Li et al., 2023b; Qi et al., 2024) and introduces a novel dual-side training enhancement. During unlearning, non-target clients use the Q-former to decouple local images into content and style features. Content features remain local and are integrated into the model via an additional linear layer, boosting classification performance and retraining efficiency. Concurrently, clients upload clustered style features to the server, which generates domain-aligned auxiliary data using stable diffusion and label prompts, effectively mitigating Domain Inconsistency through data replay.

### 4.1 STYLE-CONTENT DECOUPLING

Our inspiration comes from the fact that style features are safer than class prototypes, as they contain information such as color and texture, and it is almost impossible to restore the original image based

---

**Algorithm 1** FedSycle

---
**Input**: Q-Former, server step size $\eta_s$, client learning rate $\eta_c$, unlearning rounds $T$, local steps $K$.
**Initialize**: client model $w^o$ with linear layer for alignment, $p = (p_1, p_2, ..., p_N)$ according to local data.
**for** $t = 0$ **to** $T - 1$ **do**:
  1: **if** $t = 0$ **then**
  2:    **for** each non-target client $i$ **do**
  3:       Decouple style-content feature by the pretrained Q-former,
  4:       Do local style features clustering by Eq. 4,
  5:       Update style centroids to server.
  6:    **end for**
  7:    Server performs the server-side auxiliary data generation.
  8: **end if**
  9: Server computes $w^t_{s,k+1} = w^t_{s,k} - \eta_s g^t_{s,k}$ for each $k = 0, ..., K - 1$.
10: **for** each non-target client $i$ **do**
11:    Compute $w^t_{i,k+1} = w^t_{i,k} - \eta_c g^t_{i,k}$ for each $k = 0, ..., K - 1$.
12:    Let $\Delta^t_i = w^t_{i,K} - w^t_{i,0} = -\eta_c g^t_{i,k}$, and send it to server.
13: **end for**
14: Let $\Delta^t_s = w^t_{s,K} - w^t_{s,0} = -\eta_c g^t_{s,k}$;
15: **Server** do domain consistent aggregation as: $w^{t+1} = w^t + \alpha p_i \sum_{i=1}^N \Delta^t_i + (1 - \alpha)\Delta^t_s$;
**Output**: $w^T$ as the Federated Unlearning model $w^u$.

---

on such information. Specifically, we employ pre-trained Q-Former DeaDiff te pqi2024deadiff to decouple image content and style features. The model extracts: (1) Content features: $I_c \in \mathbb{R}^{B,H_c,D_c}$, (2) Style features: $I_s \in \mathbb{R}^{B,H_s,D_s}$, where $B$ is batch size, $H_c/H_s$ are latent representation counts, and $D_c/D_s$ are their dimensions.

**Style Feature Clustering** In order to better summarize the style features of client data, while avoiding burden on storage and communication, we perform clustering on the local style features. For each client $i$, style features $I_s$ are clustered to obtain $M$ centroids $\{c_{i,m}\}_{m=1}^M$ by solving:

$$\arg \min_{\{\mathbf{c}_{i,m}\}} \sum_{m=1}^M \sum_{\mathbf{x}_i \in \mathcal{D}^i_{\text{train}}} \| E_{w^o}(\mathbf{x}_i) - \mathbf{c}_{i,m} \|^2 \tag{4}$$

where $\mathcal{D}^i_{\text{train}}$ is the local data distribution. A style centroids set $\{c_{i,m}\}_{m=1}^M$ represents the typical style characteristic of client $i$'s local data, reduces storage and computational burdens on the server.

## 4.2 CLIENT-SIDE FAST RETRAINING

Despite sharing identical labels, data from different domains often exhibits varying learning difficulties due to domain-specific characteristics. To address this challenge, we leverage content latent features $I_c$ to enhance the model's ability to decouple semantic content from domain-specific style attributes during local retraining. We introduce a learnable alignment linear layer to combine the content features $I_c$ with the FL model's representation:

$$\begin{cases} \mathbf{F}_{content} = W_{align} \cdot \text{Flatten}(I_c) + b_{align}, \\ \mathbf{F}_{final} = \mathbf{F}_{ori} + \mathbf{F}_{content}, \\ \text{Output} = \text{Softmax}(W_{cls} \cdot \mathbf{F}_{final} + b_{cls}), \end{cases} \tag{5}$$

where $\mathbf{F}_{ori}$ denotes the feature representations extracted by the FL model's feature extraction module, $W_{align}$ and $b_{align}$ are the weights of linear alignment layer, $\mathbf{F}_{content}$ is the aligned content feature vector, and $\text{Flatten}(\cdot)$ denotes the operation that flattens the content feature into a 2D matrix. The combined feature vector $\mathbf{F}_{final}$ is fed into the classification layer, where $W_{cls}$ and $b_{cls}$ are the weights of classification layer. This feature fusion combines features from Deadiff and the FL model, which not only ensure that the FL model can effectively learn from the task-specific dataset, but also leverage the powerful generalization ability of the PTM.

## 4.3 SERVER-SIDE DOMAIN INCONSISTENCY REDUCTION

**Auxiliary Data Generation.** After receiving cluster centroids $\{c_{i,m}\}_{i \in N, m \in M}$ which denote the typical style features from the clients, we have multiple style features and a fixed number $L$ of known

labels (full set knowledge doesn't compromise the privacy of the local clients). For each style feature $c_{i,m}$, we use pre-trained stable diffusion model to generate $M \times L$ auxiliary images by combining the prompts of all label texts encoded by CLIP and styles.

On the server side, as shown in Fig. 3, we utilize the supplementary data to compensate for the Domain Inconsistency. An additional training is performed on the synthetic server auxiliary data, which is then combined with the updates from local clients through a hyper-parameter $\alpha$ as shown in the following equation:

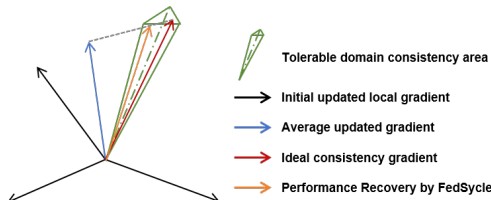

$$w^{t+1} = w^t + \alpha \frac{1}{N} \sum_{i=1}^{N} \Delta_i^t + (1-\alpha)\Delta_s^t, \quad (6)$$

Figure 3: Server-side Domain Inconsistency reduction

where $\Delta_i^t$ represents the updates by clients $i$ at communication rounds $t$. This server-step guidance can achieve recovery of Domain Inconsistency while ensuring the convergence of the federated system through leveraging the auxiliary consistent dataset generated on the server.

### 4.4 CONVERGENCE ANALYSIS OF FEDSYCLE

To provide the convergence analysis for FedSycle, following (Wang et al., 2020; 2024), we make the assumptions below:

**Assumption 1 (L-Smoothness)** *Each objective function of clients is Lipschitz smooth, that is, there exists a constant $L > 0$, such that $\|\nabla F_i(\boldsymbol{x}) - \nabla F_i(\boldsymbol{y})\| \leq L \|\boldsymbol{x} - \boldsymbol{y}\|, \forall i \in \{1, 2, \ldots, N\}$.*

**Assumption 2 (Unbiased Gradient and Bounded Variance)** *The stochastic gradient calculated by each client can be considered as an unbiased estimator of the clients' gradient $\mathbb{E}_\xi[g_i(w|\xi)] = \nabla F_i(w)$, and has bounded variance $\mathbb{E}_\xi[\|g_i(w|\xi) - \nabla F_i(w)\|^2] \leq \sigma^2, \forall i \in \{1, \ldots, N\}, \sigma^2 \geq 0$.*

**Assumption 3 (Bounded Dissimilarity of Clients' Gradient)** *For any sets of weights $\{p_i^t \geq 0\}_{i=1}^N, \sum_{i=1}^N p_i^t = 1$,, there exist constants $(\gamma^2 + 1) \geq 1$, $A^2 \geq 1$, such that $\sum_{i=1}^N p_i^t \|\nabla F_i(w)\|^2 \leq \gamma^2 \left\|\sum_{i=1}^N p_i^t \nabla F_i(w)\right\|^2 + A^2$.*

**Theorem 1 (Convergence Analysis of FedSycle)** *Under Assumptions 1-3, the unlearning rounds T is pre-determined. Let $\eta_s$, $\eta_c$ be the server and client step size. If $\eta_c$ is small enough such that $\eta_c < \min\left(\frac{1}{8LK}, C\right)$, where $\frac{1}{2} - 20L^2 \frac{1}{N} \sum_{i=1}^N K^2 \eta_L^2 (\gamma^2 + 1)^2 > C > 0$ and $\eta \leq \frac{1}{\eta_s L}$, then with the proper setting $\eta_c = \mathcal{O}\left(\frac{1}{\sqrt{T}KL}\right)$ and $\eta_s = \mathcal{O}\left(\sqrt{NK}\right)$ the convergence rate is:*

$$\min_{t \in [T]} \mathbb{E} \|\nabla f(w_t)\|^2 \leq \mathcal{O}\left(\sqrt{\frac{N}{KT}} + \frac{1}{T}\right). \quad (7)$$

The details of convergence analysis and discussion can be found in Appendix A.

## 5 EXPERIMENTS

### 5.1 EXPERIMENTAL SETUP

**Datasets and Models.** We evaluate the performance of the complex classification task on two typical domain image datasets to assess the performance of FedSycle.

- PACS dataset, consisting of images from 4 distinct domains: photo, art painting, cartoon, and sketch with the same 7 labels.
- DomainNet dataset, consisting of images from 6 distinct domains: clipart, infograph, painting, quickdraw, real, and sketch with the same 345 labels. Following (Yang et al., 2024) we randomly pick 30 classes with three different random seeds.

Following (Pan et al., 2023; Tam et al., 2024; Wang et al., 2023), we adopt three different Non-IID data partitioning methods (range from extreme to conventional and almost cover possible application scenarios) as:

- One client owns the complete data of one domain.
- Par-2 allocation divides each domain's data into two shards, with each client selecting two.
- Dirichlet data partitioning with Non-IID parameter $\alpha$.

(a) Results on one client one domain data partition, running on MobileNetV2.

| Method | DomainNet | | | PACS | | |
|---|---|---|---|---|---|---|
| | NT Acc. ↑ | ASR ↓ | DI ↓ | NT Acc. ↑ | ASR ↓ | DI ↓ |
| Pretrain | 43.24±2.71 | 39.23±1.59 | – | 45.38±0.48 | 50.63±1.55 | – |
| Retrain | 61.36±0.82 | – | – | 57.37±1.99 | – | – |
| FedEraser | 35.44±4.95 | **1.20±0.22** | 0.17±0.04 | 44.11±1.45 | 17.40±2.82 | 0.59±0.18 |
| FedRecovery | 53.41±1.14 | 7.64±1.02 | **0.05±0.01** | 48.73±3.11 | 26.16±2.10 | 0.60±0.14 |
| FedKDU | 61.09±0.27 | 3.90±0.40 | 0.06±0.02 | 66.59±2.98 | 17.22±1.69 | 0.14±0.06 |
| EWCSGA | 49.94±2.36 | 10.10±0.80 | 0.26±0.04 | 36.84±0.59 | 30.12±1.47 | 0.92±0.28 |
| MoDe | 54.34±0.41 | 5.21±0.52 | 0.06±0.01 | 58.64±3.31 | 18.55±1.82 | 0.50±0.15 |
| FedSycle | **74.70±1.51** | 5.01±0.49 | **0.05±0.01** | **83.85±2.21** | 13.17±2.49 | **0.03±0.01** |

(b) Results on Par-2 data partition, running on VGG-16.

| Method | DomainNet | | | PACS | | |
|---|---|---|---|---|---|---|
| | NT Acc. ↑ | ASR ↓ | DI ↓ | NT Acc. ↑ | ASR ↓ | DI ↓ |
| Pretrain | 73.23±1.02 | 99.71±0.44 | – | 91.84±0.15 | 99.89±0.04 | – |
| Retrain | 72.07±0.73 | – | – | 91.96±0.66 | – | – |
| FedEraser | 69.42±1.09 | **0.71±0.21** | 0.92±0.14 | 90.67±0.33 | **0.85±0.14** | 0.90±0.10 |
| FedRecovery | 73.74±1.40 | 64.95±2.76 | 1.44±0.37 | 73.74±3.35 | 46.20±1.84 | 1.08±0.22 |
| FedKDU | 69.11±0.64 | 14.72±1.18 | 0.35±0.14 | 89.69±0.92 | 1.16±0.14 | **0.11±0.02** |
| EWCSGA | 74.93±0.68 | 56.93±1.72 | 0.35±0.07 | 92.85±0.20 | 74.60±2.28 | 0.48±0.18 |
| MoDe | 74.71±0.55 | 21.98±1.94 | 0.40±0.09 | 92.63±0.15 | 16.49±1.19 | 1.28±0.22 |
| FedSycle | **86.74±1.10** | 5.13±1.22 | **0.12±0.03** | **94.47±0.16** | 1.10±0.36 | 0.17±0.04 |

Table 1: Numerical Results of FU Algorithms. The best-performing algorithm is indicated in **bold**, and the second-best is shown in blue. (a) We designate the client possessing real (cartoon) domain data as the target client for DomainNet (PACS) dataset. (b) We designate the client possessing real and sketch (cartoon and photo) domain data as the target client for DomainNet (PACS) dataset.

We evaluate three models: ResNet-18 (He et al., 2016), MobilenetV2 (Sandler et al., 2018), and VGG16 (Simonyan, 2014) and report the more challenging tasks and primary results in the main paper, with additional results available in Appendix B.

**Baselines and Metrics.** We first perform retraining from scratch to serve as a standard for evaluating various FU algorithms, and consider various types of mainstream FU algorithms, including FedEraser (Liu et al., 2021), FedRecovery (Zhang et al., 2023), FedKDU (Wu et al., 2022a), EWCSGA (Wu et al., 2022b), MoDe (Zhao et al., 2023b) and KNOT (Su & Li, 2023) (Appendix B). They are representative algorithms based on historical information, reverse learning, and other approaches, respectively. We use attack success rate, non-target accuracy (Liu et al., 2021; Wu et al., 2022a;b; Zhang et al., 2023; Zhao et al., 2023b), Domain Inconsistency, and unlearn efficiency for algorithm evaluation.

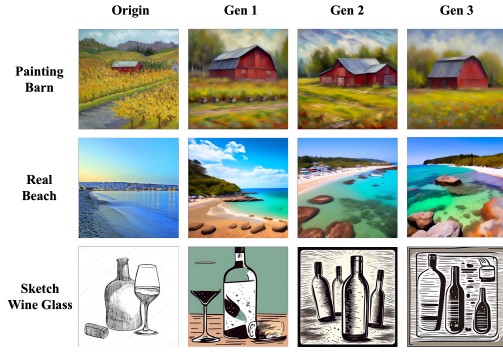

Figure 4: Comparison between dataset images and generated images. It's hard to reconstruct the images.

- Attack success rate (ASR) is the metric to evaluate effectiveness of an unlearn algorithm, which insert a backdoor trigger to mark the target data to accurately identify target data during both training and unlearning processes. Lower the ASR, better the unlearn effectiveness.
- Non-target clients' acccuracy (NT Acc.): We conduct utility tests on the non-target clients that remain within the FL system and report the average performance as the non-target accuracy (NT Acc.). Higher the NT Acc., better the utility of the global model.
- Domain Inconsistency (DI) (Def. 1). Lower the DI, better model unbiasedness.
- Unlearn Rounds. We use the convergence rounds as evidence of unlearning efficiency. The Shorter the convergence rounds, higher the efficiency.

We run all experiments on NVIDIA GEFORCE 3090 (24GB) with 100 target unlearning rounds and 30 performance recovery rounds, a local epoch $E = 1$, using SGD as the optimizer of batch size $B = 128$, and learning rate $lr = 0.01$ with a decay of 0.998 per communication round. We report the best test results of all algorithm averaged over three random seeds. We randomly (without loss of generality) select a client as the target client for unlearning and report the results in the main paper. We illustrate the generated examples in Fig. 4. Other circumstances are reported in Appendix B.

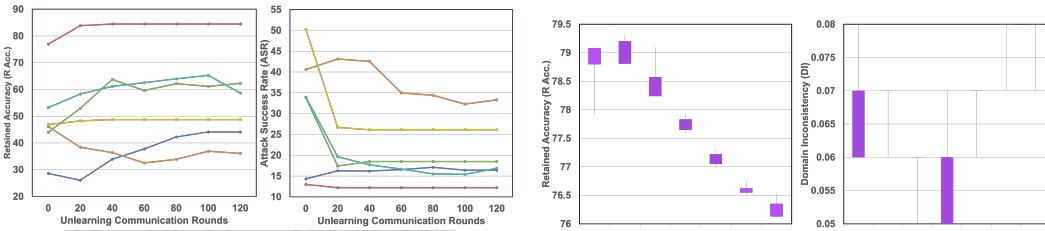

(a) Performance of Different Algorithms in Retrain Clients Accuracy (NT Acc.) and Attack Success Rate (ASR) based on one client one domain PACS.

(b) Hyper-parameter $\alpha$ experiments of FedSycle based on Pat-2 DomainNet task running with MobileNetV2.

Figure 5: Illustration of the convergence efficiency and stability of FedSycle.

## 5.2 NUMERICAL RESULTS

(a) Performance Comparison of Algorithms. As shown in Tab. 1 and Fig. 5a, FedSycle achieves breakthrough results with style-content decoupling. Specifically:

- **Dominant Model Generalization Performance.** FedSycle meets the unlearning effectiveness almost from the outset, as it is fundamentally a retraining-based method, and it is capable of restoring the model's utility within the shortest communication rounds. Numerically, FedSycle achieves at least **31%** higher non-target accuracy (NT Acc.) compared to the average performance of baselines across all tasks. It significantly outperforms the second-best algorithm by **43%** on the PACS (MobileNetV2) task and even achieves a **2%** breakthrough on the nearly saturated PACS (VGG16) task. This success is due to the style-content decoupling, which enables the model to better capture core content features for label classification, thereby surpassing previous performance limits.
- **Significant Reduction in Domain Inconsistency.** Thanks to the server-side recovery, generation, and replay of style features, FedSycle achieves stable and lower Domain Inconsistency (DI). It outperforms the average of all baselines by **83%** across all tasks. The server-side recovery make up for the inherent domain heterogeneity of the original system and any new heterogeneity possibly introduced by FU.
- **Fast Convergence.** As shown in Fig. 5a. FedSycle complete the convergence of the FU algorithm in the shortest communication round, demonstrating the excellent convergence efficiency of the algorithm. We further discuss this in Appendix

(b) **Ablation Study**.

- We conduct ablation experiments on FedSycle's client-side fast retraining and server-side Domain Inconsistency reduction based on retraining from scratch. The results in Tab. 2 indicate that the improvement in non-target accuracy primarily stems from feature decoupling on the client side and the direct addition of content features to training. In contrast, the improvement in DI is mainly attributed to the supplementary data generated by server-side style features, which balances the domain distribution.

| Method | NT Acc.↑ | ASR↓ | DI↓ |
|---|---|---|---|
| Baselines Avg. | 50.84 | 5.60 | 0.12 |
| Retrain | 61.36 | – | – |
| + ① client-side | 75.52 | 4.91 | 0.08 |
| + ② server-side | 73.91 | 5.23 | 0.06 |
| + ① ② FedSycle | 74.70 | 5.01 | 0.05 |

Table 2: Dual-Side Ablation Study of FedSycle (one client on domain partition DomainNet task on MobileNetV2)

- We evaluate the impact of different hyperparameters $\alpha$ on FedSycle's performance, as illustrated in Fig. 5b, which indicates that FedSycle is robust to hyperparameter $\alpha$ and does not require fine-grained fine-tuning.

## 6 CONCLUSION LIMITATION AND FUTURE WORKS

Our federated unlearning algorithm leverages style-content decoupling to effectively unlearn, mitigate Domain Inconsistency, and preserve model utility. Experiments confirm this method rapidly restores model performance and stabilizes results by addressing data heterogeneity. Although computational constraints limited our exploration of larger models, the clear benefits in performance and domain adaptation justify our future work. We will continue exploring feature decoupling in federated learning to reduce the negative impacts of data heterogeneity.

# STATEMENTS

## ETHICS STATEMENT

The authors of this paper have read and agree to abide by the ICLR Code of Ethics. We believe that this work does not raise any significant ethical concerns. Our research did not involve experiments with human subjects, nor did it process sensitive personal data. All datasets used in our study are publicly available. We foresee no direct negative societal impacts from the methods and potential applications presented in this work.

## REPRODUCIBILITY STATEMENT

We are committed to ensuring the reproducibility of our research. We have provided comprehensive experimental details in the main paper and Appendix B, including dataset preprocessing procedures, model architecture specifications, full training details, and all hyperparameter configurations. Furthermore, we will make our source code and model checkpoints publicly available.

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

# A PROOF OF THEOREM 1: CONVERGENCE ANALYSIS OF FEDSYCLE

Recall the assumptions we've made to derivate the convergence of FedSycle:

**Assumption 1 (Smoothness)** *Each objective function of clients is Lipschitz smooth, that is, there exists a constant $L > 0$, such that $\|\nabla F_i(\boldsymbol{x}) - \nabla F_i(\boldsymbol{y})\| \leq L\|\boldsymbol{x} - \boldsymbol{y}\|, \forall i \in \{1, 2, \ldots, N\}$.*

**Assumption 2 (Unbiased Gradient and Bounded Variance)** *The stochastic gradient calculated by each client can be an unbiased estimator of the clients' gradient $\mathbb{E}_\xi[g_i(\boldsymbol{w}|\xi)] = \nabla F_i(\boldsymbol{w})$, and has bounded variance $\mathbb{E}_\xi[\|g_i(\boldsymbol{w}|\xi) - \nabla F_i(\boldsymbol{w})\|^2] \leq \sigma^2, \forall i \in \{1, 2, \ldots, N\}, \sigma^2 \geq 0$. We assume the server's stochastic gradient $g_s$ is also unbiased with variance bounded by $\sigma_s^2$.*

**Assumption 3 (Bounded Dissimilarity of Clients' Gradient)** *For any sets of weights $\{p_i^t \geq 0\}_{i=1}^N, \sum_{i=1}^N p_i^t = 1,$, there exist constants $\gamma \geq 1$, $A \geq 0$, such that $\sum_{i=1}^N p_i^t \|\nabla F_i(\boldsymbol{w})\|^2 \leq \gamma^2 \left\|\sum_{i=1}^N p_i^t \nabla F_i(\boldsymbol{w})\right\|^2 + A^2$.*

When $\alpha \neq 0$, we make an additional assumption to bound differences between the server guidance gradients and the updated gradients as below:

**Assumption 4** *We assume the difference between the server gradient $\nabla F_s(\boldsymbol{w})$ and the global gradient $\nabla f(\boldsymbol{w})$ is bounded by the magnitude of the global gradient itself. Specifically, there exists a constant $\beta \geq 0$ such that for all $t$:*

$$\mathbb{E}\left[\|\nabla F_s(\boldsymbol{w}^t) - \nabla f(\boldsymbol{w}^t)\|^2\right] \leq \beta\|\nabla f(\boldsymbol{w}^t)\|^2$$

## A.1 PRELIMINARIES

For the ease of writing, we define the following auxiliary variables:

$$\text{Normalized Stochastic Gradient:} \quad \boldsymbol{d}_i^{(t)} = \sum_{k=0}^{K-1}[\alpha g_i(\boldsymbol{w}_i^{t,k}) + (1-\alpha)g_s(\boldsymbol{w}^t))], \tag{8}$$

$$\text{Normalized Gradient:} \quad \boldsymbol{h}_i^{(t)} = \sum_{k=0}^{K-1}[\alpha \nabla F_i(\boldsymbol{w}_i^{t,k}) + (1-\alpha)\nabla F_s(\boldsymbol{w}^t))]. \tag{9}$$

Note that the expectation of the stochastic gradient is taken with respect to the client's local data $\xi_i$ and server's data $\xi_s$. So, $\mathbb{E}[\boldsymbol{d}_i^{(t)}] = \boldsymbol{h}_i^{(t)}$.

## CONVERGENCE ANALYSIS WITH NEW ASSUMPTION

## A.2 INITIAL SETUP

From the L-smoothness of $f(\boldsymbol{w}) = \sum_{i=1}^N p_i F_i(\boldsymbol{w})$, we have:

$$\mathbb{E}\left[f(\boldsymbol{w}^{t+1})\right] - f(\boldsymbol{w}^t) \leq -\eta \underbrace{\mathbb{E}\left[\left\langle \nabla f(\boldsymbol{w}^t), \sum_{i=1}^N p_i \boldsymbol{d}_i^{(t)}\right\rangle\right]}_{T_1} + \frac{\eta^2 L}{2}\underbrace{\mathbb{E}\left[\left\|\sum_{i=1}^N p_i \boldsymbol{d}_i^{(t)}\right\|^2\right]}_{T_2}. \tag{10}$$

We first bound the $T_1$ term. Since $\mathbb{E}[\boldsymbol{d}_i^{(t)}] = \boldsymbol{h}_i^{(t)}$, we have:

$$T_1 = \mathbb{E}\left[\left\langle \nabla f(\boldsymbol{w}^t), \sum_{i=1}^N p_i \boldsymbol{h}_i^{(t)}\right\rangle\right] \tag{11}$$

$$= \frac{1}{2}\mathbb{E}\left[\|\nabla f(\boldsymbol{w}^t)\|^2 + \left\|\sum_{i=1}^N p_i \boldsymbol{h}_i^{(t)}\right\|^2 - \left\|\nabla f(\boldsymbol{w}^t) - \sum_{i=1}^N p_i \boldsymbol{h}_i^{(t)}\right\|^2\right]. \tag{12}$$

For $T_2$, we have:

$$T_2 = \mathbb{E}\left[\left\|\sum_{i=1}^{N} p_i\left(\boldsymbol{d}_i^{(t)} - \boldsymbol{h}_i^{(t)}\right) + \sum_{i=1}^{N} p_i\boldsymbol{h}_i^{(t)}\right\|^2\right] \tag{13}$$

$$\leq 2\mathbb{E}\left[\left\|\sum_{i=1}^{N} p_i\left(\boldsymbol{d}_i^{(t)} - \boldsymbol{h}_i^{(t)}\right)\right\|^2\right] + 2\mathbb{E}\left[\left\|\sum_{i=1}^{N} p_i\boldsymbol{h}_i^{(t)}\right\|^2\right] \tag{14}$$

$$= 2\sum_{i=1}^{N} p_i^2\mathbb{E}\left[\left\|\boldsymbol{d}_i^{(t)} - \boldsymbol{h}_i^{(t)}\right\|^2\right] + 2\mathbb{E}\left[\left\|\sum_{i=1}^{N} p_i\boldsymbol{h}_i^{(t)}\right\|^2\right]. \tag{15}$$

The variance term can be bounded as:

$$\mathbb{E}[\|\boldsymbol{d}_i^{(t)} - \boldsymbol{h}_i^{(t)}\|^2] = \mathbb{E}\left[\left\|\sum_{k=0}^{K-1} \alpha(g_i(\boldsymbol{w}_i^{t,k}) - \nabla F_i(\boldsymbol{w}_i^{t,k}))\right\|^2\right] \leq K\alpha^2\sigma^2. \tag{16}$$

Assuming $p_i = 1/N$, $\sum p_i^2 = 1/N$. So $T_2 \leq \frac{2K\alpha^2\sigma^2}{N} + 2\mathbb{E}[\|\sum p_i\boldsymbol{h}_i^{(t)}\|^2]$.

Plugging $T_1$ and $T_2$ into (10):

$$\mathbb{E}[f(\boldsymbol{w}^{t+1})] - f(\boldsymbol{w}^t) \leq -\frac{\eta}{2}\|\nabla f(\boldsymbol{w}^t)\|^2 + \frac{\eta^2 LK\alpha^2\sigma^2}{N}$$
$$- \frac{\eta}{2}(1 - 2\eta L)\mathbb{E}\left[\left\|\sum p_i\boldsymbol{h}_i^{(t)}\right\|^2\right] + \frac{\eta}{2}\mathbb{E}\left[\left\|\nabla f(\boldsymbol{w}^t) - \sum p_i\boldsymbol{h}_i^{(t)}\right\|^2\right]. \tag{17}$$

By setting $\eta L \leq 1/4$, the term with $\mathbb{E}[\|\sum p_i\boldsymbol{h}_i^{(t)}\|^2]$ is non-positive and can be dropped:

$$\mathbb{E}[f(\boldsymbol{w}^{t+1})] - f(\boldsymbol{w}^t) \leq -\frac{\eta}{2}\|\nabla f(\boldsymbol{w}^t)\|^2 + \frac{\eta^2 LK\alpha^2\sigma^2}{N} + \frac{\eta}{2}\underbrace{\mathbb{E}\left[\left\|\nabla f(\boldsymbol{w}^t) - \sum p_i\boldsymbol{h}_i^{(t)}\right\|^2\right]}_{T_3}. \tag{18}$$

We now bound the term $T_3$.

Let $\bar{\boldsymbol{h}}^{(t)} = \sum_i p_i\boldsymbol{h}_i^{(t)}$. We bound $T_3$ as follows:

$$T_3 = \mathbb{E}\left[\left\|\nabla f(\boldsymbol{w}^t) - \bar{\boldsymbol{h}}^{(t)}\right\|^2\right]$$

$$= \mathbb{E}\left[\left\|\sum_{k=0}^{K-1} \alpha(\nabla f(\boldsymbol{w}^t) - \sum_i p_i\nabla F_i(\boldsymbol{w}_i^{t,k})) + K(1-\alpha)(\nabla f(\boldsymbol{w}^t) - \nabla F_s(\boldsymbol{w}^t))\right\|^2\right]$$

$$\leq 2\mathbb{E}\left[\left\|\alpha\sum_{k=0}^{K-1}(\nabla f(\boldsymbol{w}^t) - \sum_i p_i\nabla F_i(\boldsymbol{w}_i^{t,k}))\right\|^2\right] + 2K^2(1-\alpha)^2\mathbb{E}\left[\|\nabla f(\boldsymbol{w}^t) - \nabla F_s(\boldsymbol{w}^t)\|^2\right]$$

$$\leq 2\alpha^2 K\sum_{k=0}^{K-1}\mathbb{E}\left[\left\|\sum_i p_i(\nabla F_i(\boldsymbol{w}^t) - \nabla F_i(\boldsymbol{w}_i^{t,k}))\right\|^2\right] + 2K^2(1-\alpha)^2\beta\|\nabla f(\boldsymbol{w}^t)\|^2 \tag{19}$$

$$\leq 2\alpha^2 KL^2\sum_{k=0}^{K-1}\sum_i p_i\mathbb{E}\left[\left\|\boldsymbol{w}^t - \boldsymbol{w}_i^{t,k}\right\|^2\right] + 2K^2(1-\alpha)^2\beta\|\nabla f(\boldsymbol{w}^t)\|^2. $$

The client drift $\mathbb{E}[\|\boldsymbol{w}^t - \boldsymbol{w}_i^{t,k}\|^2]$ can be bounded by (using standard analysis):

$$\mathbb{E}[\|\boldsymbol{w}^t - \boldsymbol{w}_i^{t,k}\|^2] \leq \eta^2 K^2(2\sigma^2 + 4\gamma^2\|\nabla f(\boldsymbol{w}^t)\|^2 + 4A^2). \tag{20}$$

Plugging this back into the bound for $T_3$:

$$T_3 \leq 2\alpha^2 KL^2(K \cdot \eta^2 K^2(2\sigma^2 + 4\gamma^2 \|\nabla f(\boldsymbol{w}^t)\|^2 + 4A^2)) + 2K^2(1-\alpha)^2\beta\|\nabla f(\boldsymbol{w}^t)\|^2$$
$$= \left(8\alpha^2 K^4\eta^2 L^2\gamma^2 + 2K^2(1-\alpha)^2\beta\right)\|\nabla f(\boldsymbol{w}^t)\|^2 + 4\alpha^2 K^4\eta^2 L^2(2\sigma^2 + 4A^2). \quad (21)$$

Now, we substitute this bound for $T_3$ into inequality (18):

$$\mathbb{E}[f(\boldsymbol{w}^{t+1})] - f(\boldsymbol{w}^t) \leq -\frac{\eta}{2}\|\nabla f(\boldsymbol{w}^t)\|^2 + \frac{\eta^2 LK\alpha^2\sigma^2}{N}$$
$$+ \frac{\eta}{2}\left[\left(8\alpha^2 K^4\eta^2 L^2\gamma^2 + 2K^2(1-\alpha)^2\beta\right)\|\nabla f(\boldsymbol{w}^t)\|^2 + 4\alpha^2 K^4\eta^2 L^2(2\sigma^2 + 4A^2)\right]$$
$$= -\frac{\eta}{2}\left(1 - 2K^2(1-\alpha)^2\beta - 8\alpha^2 K^4\eta^2 L^2\gamma^2\right)\|\nabla f(\boldsymbol{w}^t)\|^2$$
$$+ \frac{\eta^2 LK\alpha^2\sigma^2}{N} + 2\eta^3\alpha^2 K^4 L^2(2\sigma^2 + 4A^2). \quad (22)$$

For convergence, we need the coefficient of $\|\nabla f(\boldsymbol{w}^t)\|^2$ to be positive. First, we require $1 - 2K^2(1-\alpha)^2\beta > 0$, which implies $\beta < \frac{1}{2K^2(1-\alpha)^2}$. Let's define $C_\beta = 1 - 2K^2(1-\alpha)^2\beta > 0$.

Next, we choose a step size $\eta$ small enough such that $8\alpha^2 K^4\eta^2 L^2\gamma^2 \leq C_\beta/2$. This ensures that $C_\beta - 8\alpha^2 K^4\eta^2 L^2\gamma^2 \geq C_\beta/2$. With this choice, the inequality becomes:

$$\mathbb{E}[f(\boldsymbol{w}^{t+1})] - f(\boldsymbol{w}^t) \leq -\frac{\eta C_\beta}{4}\|\nabla f(\boldsymbol{w}^t)\|^2 + \eta^2\frac{LK\alpha^2\sigma^2}{N} + 2\eta^3\alpha^2 K^4 L^2(2\sigma^2 + 4A^2). \quad (23)$$

Rearranging the terms, we get:

$$\frac{\eta C_\beta}{4}\|\nabla f(\boldsymbol{w}^t)\|^2 \leq f(\boldsymbol{w}^t) - \mathbb{E}[f(\boldsymbol{w}^{t+1})] + \eta^2\frac{LK\alpha^2\sigma^2}{N} + 2\eta^3\alpha^2 K^4 L^2(2\sigma^2 + 4A^2). \quad (24)$$

Summing from $t = 0$ to $T - 1$, taking the average over $T$, and using the telescoping sum property on $f(\boldsymbol{w}^t) - \mathbb{E}[f(\boldsymbol{w}^{t+1})]$:

$$\frac{1}{T}\sum_{t=0}^{T-1}\mathbb{E}[\|\nabla f(\boldsymbol{w}^t)\|^2] \leq \frac{4}{\eta T C_\beta}\sum_{t=0}^{T-1}(f(\boldsymbol{w}^t) - \mathbb{E}[f(\boldsymbol{w}^{t+1})]) + \frac{4\eta LK\alpha^2\sigma^2}{NC_\beta} + \frac{8\eta^2\alpha^2 K^4 L^2(2\sigma^2 + 4A^2)}{C_\beta}$$
$$\leq \frac{4(f(\boldsymbol{w}^0) - f^*)}{\eta T C_\beta} + \frac{4\eta LK\alpha^2\sigma^2}{NC_\beta} + \frac{8\eta^2\alpha^2 K^4 L^2(2\sigma^2 + 4A^2)}{C_\beta}. \quad (25)$$

where $f^*$ is the minimum value of $f(\boldsymbol{w})$.

We summarize all learning rate constraint as

$$\eta L \leq \frac{1}{2K}, \quad 4\eta^2 L^2 K(K-1) \leq \frac{1}{2\gamma^2 + 1}. \quad (26)$$

By setting $\eta = \sqrt{\frac{N}{KT}}$, we have

$$\min_{t\in[T]}\mathbb{E}\|\nabla f(\boldsymbol{w}_t)\|^2 \leq \mathcal{O}\left(\sqrt{\frac{N}{KT}} + \frac{1}{T}\right). \quad (27)$$

# B  EXPERIMENTAL DETAILS AND SUPPLEMENTARY RESULTS

## B.1  DOMAIN ANALYSIS

In a FL environment involving domain heterogeneity, data from the same domain may be held by different clients, while different clients may hold data with the same labels but from different domains. As illustrated in Fig 6, such data distribution leads to the coupling of domain feature spaces, which may be the reason for the domain performance degradation and domain inconsistency observed during unlearning. What's more, due to this coupling of domains, unlearning one domain might make learning another domain easier. Therefore, we point out that inconsistency can be caused by either an improvement or a decline in domain performance, or both, which is also shown in Fig. 7.

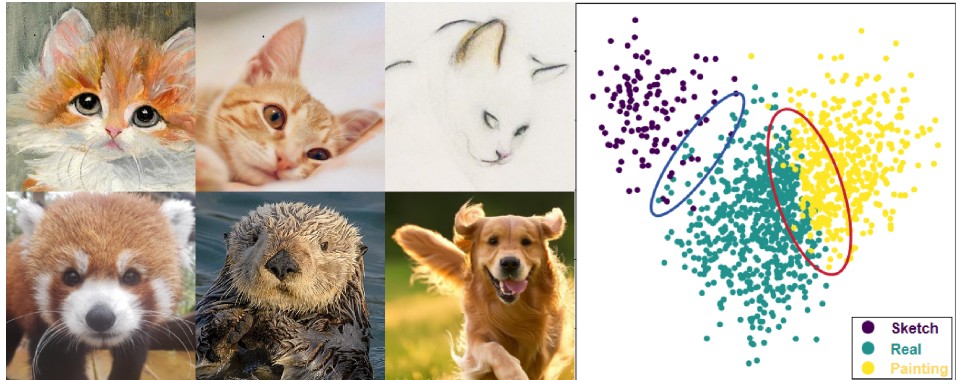

Figure 6: The image feature oracle of FL. FL model can effectively learn data features from different domains and class labels. However, within the same label but different domains, and within the same domain but different labels, there is feature coupling and can easily lead to domain inconsistency when executing FU.

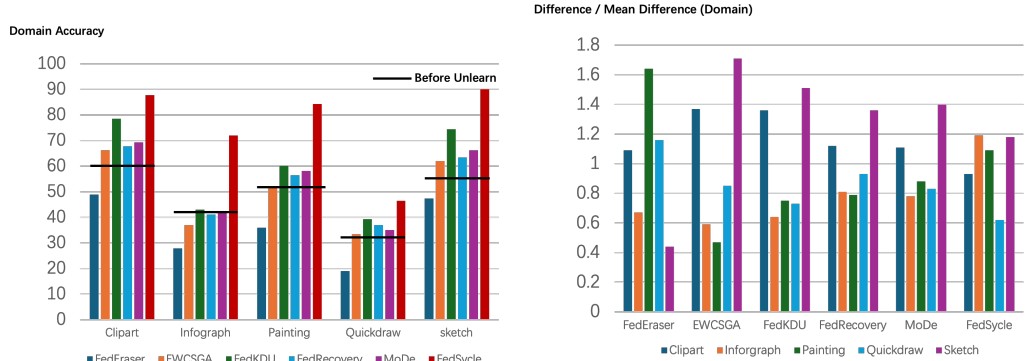

Figure 7: The leaving client (target client) may have conflicts in data domain and labels with other clients remaining in the FL system. Unlearning the target client can affect the performance of other domains, and the extent of the impact varies across domains, leading to domain inconsistency.

## B.2 EXPERIMENTAL SETUP

**Number and selection of Clients.** Based on the cross-silo scenario, without loss of generality, we select the same number of clients as the dataset domains. Specifically, for the DomainNet-related experiments, the number of clients is 6, and for the PACS-related experiments, the number of clients is 4.

## B.3 HYPER-PARAMETERS

We follow the hyperparameter settings of the baselines or the default settings of the source code for hyperparameter tuning. The specific hyperparameters involved for each algorithm and the tuning range are shown in the Tab. 3. Our results are averaged over three random seeds, and the best set of hyperparameters for each algorithm is reported.

## B.4 MODEL STRUCTURES

As shown in Fig. 8, we present the complete structure of the used pretrained Q-former named "Deadiff" Qi et al. (2024). The image is transformed into latent features through pretrained encoder. We prompt the pre-trained Q-former from DeaDiff with 'content' and 'style' text inputs to specify which parts of the image need to be extracted. The extraction process begins by initializing a 16*768 tensor and generating a new tensor combining the input image's latent features and the prompt. This new tensor contains the image's content or style features (depending on the prompt). We upload the locally extracted style features to the server, and on the server side, the uploaded style is combined with the target class label's prompt as a condition to instruct the stable diffusion model to generate

| Algorithm | Hyper-parameters |
|---|---|
| FedEraser | $\Delta_t = 5$ |
| FedKDU | $\tau \in \{0.5, 1.0, 2.0, 3.0\}, \alpha \in \{0.1, 0.3, 0.5\}$ |
| MoDe | $\lambda \in \{0.3, 0.5, 0.7\}, r_{de} \in \{0.5 \times T, 0.7 \times T\}$ |
| KNOT | Cluster_Num=2 |
| EWCSGA | $\lambda \in \{0.3, 0.5, 0.7\}, gradient\_clip \in \{1.0, 3.0, 5.0, 10.0\}$ |
| FedSycle | $\alpha \in \{0.0, 0.1, 0.3, 0.5, 0.7, 0.9, 1.0\}$ |

Table 3: Hyperparameters of all algorithms.

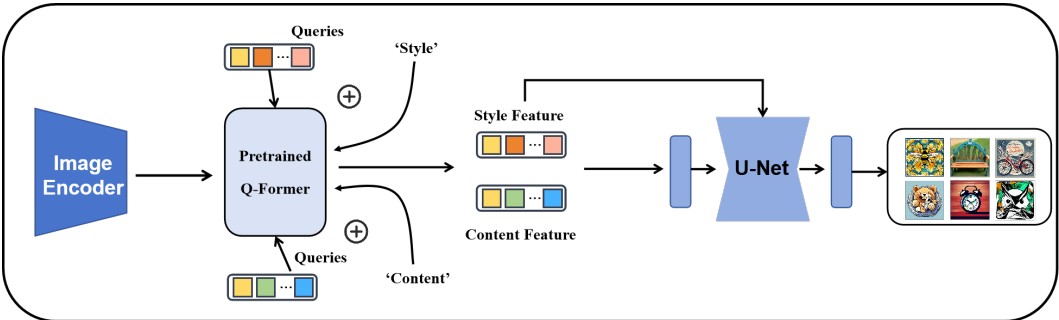

Figure 8: Structures of Style-Content Decoupling and Auxiliary Data Generation.

auxiliary images. In this paper, we use pretrained Stable Diffusion v1.5 with an unconditional guidance scale of 7.5.

### B.5 PROMPTS USED FOR DATA GENERATION

System Conditional filter: "over-exposure, under-exposure, saturated, duplicate, out of frame, lowres, cropped, worst quality, low quality, jpeg artifacts, morbid, mutilated, out of frame, ugly, bad anatomy, bad proportions, deformed, blurry, duplicate"

Prompt Template: A clear and easily identifiable {label}.

Figure 9: Prompt Template.

We use simple and clear instruction as shown in Fig 9 to prompt the diffusion model to generate the server-side auxiliary dataset.

### B.6 CLAIM ON COST OF FEDSYCLE

We conduct a evaluation on DomainNet (MobileNet-V2) of the extra cost based on our NVIDIA GEFORCE RTX 3090 (24GB) computational power, which includes style-content decoupling, auxiliary image dataset generation, guidance gradients computing, and local training. As shown in Tab. 4:

- The style-content decoupling operation for each image takes an average of 0.016 seconds.
- Generating each auxiliary image requires an average of 4 seconds.
- Clients require an additional 7GB of space for style-content decoupling.
- Each style-content features needs only 96kb to store locally.

The model takes 0.016 seconds to do style-content decoupling for each image and 4 seconds to generate each auxiliary image. This does not pose a burden on the federated learning system. It is important to note that the decoupling of images and the generation of auxiliary images are one-time processes. Clients can save the content features corresponding to each image for subsequent training (only 96kb per image), thus avoiding repeated time consumption.

However, we emphasize that this is **on the same order of magnitude as the time consumption** of a single training round of FL. The additional time consumption per round leads to significantly better performance and lower domain inconsistency.

|  | Extra i | Extra ii | Extra iii | Extra iv |
|---|---|---|---|---|
| Time | 0.016s/img | 4s/img | 41.10s (+0s) | 55.23s (+14.13s First Round) |
| Memory | 7GB | 10.59GB | 1.41MB | 1.41MB |

Table 4: Extra Cost of FedSycle: i. Extra time for style-content feature decoupling; ii. Extra time for auxiliary images generation; iii. Extra time for server guidance gradients computing; iv. Extra time for local training.

| Time (mins) | FedEraser | FedRecovery | FedKDU | EWCSGA | MoDe | FedSycle |
|---|---|---|---|---|---|---|
| 20 | 31.71 / 0.15 / 3.50 | 51.98 / 0.07 / 8.83 | 51.97 / 0.08 / 7.10 | 40.46 / 0.45 / 26.20 | 52.23 / 0.06 / 5.61 | 71.37 / 0.09 / 3.00 |
| 40 | 37.75 / 0.17 / 1.23 | 54.04 / 0.05 / 6.39 | 61.33 / 0.06 / 4.10 | 41.55 / 0.17 / 20.07 | 53.35 / 0.05 / 5.20 | 74.62 / 0.05 / 2.10 |
| 60 | - | - | - | 42.51 / 0.26 / 17.70 | 54.24 / 0.05 / 5.54 | 76.18 / 0.05 / 2.13 |
| 80 | - | - | - | 49.99 / 0.24 / 10.20 | 54.41 / 0.05 / 5.32 | - |
| Converge | 37.75 / 0.17 / **1.23** | 54.04 / **0.05** / 6.39 | 61.33 / **0.06** / 4.10 | 49.99 / 0.24 / 10.20 | 54.41 / **0.05** / 5.32 | **76.18** / **0.05** / 2.13 |

Table 5: Convergence Efficiency Comparison with Baselines. This is the milestone result of training DomainNet using MobileNetV2 under the one client one domain setting. For fairness, we use the algorithm running time as the measurement. At each milestone moment, we report the NT Acc. / DI / ASR under each algorithm. The red box represents that FedSycle first exceeded all baselines in a specific indicator at that moment. We also list the performance when the model has fully converged. The best performance is **bolded**, and the second best is marked in blue.

Due to the uniqueness of federated unlearning, the communication steps and frequencies of different algorithms may vary. It is difficult to uniformly and fairly compare the convergence efficiency through communication rounds. Therefore, we use a more direct approach to compare the convergence situation, that is, the algorithm convergence time. As shown in Tab. **??**, in the early stage of training, our method quickly gains a leading position in both NT Acc. and ASR metrics. And at the second milestone, it achieves the lowest DI. After that, FedSycle maintains the highest NT Acc. and the lowest DI, indicating that although FedSycle introduces additional model and training parameters, the convergence efficiency and performance of the algorithm have significantly improved, which can be considered a reasonable trade-off.

## B.7 ADDITIONAL NUMERICAL RESULTS

(a) Results on one client one domain data partition, running on MobileNetV2.

| Method | DomainNet | | | | PACS | | | |
|---|---|---|---|---|---|---|---|---|
|  | NT Acc. | ASR | DI | Efficiency | NT Acc. | ASR | DI | Efficiency |
| Pretrain | 43.24±2.71 | 39.23±1.59 | - | - | 45.38±0.48 | 50.63±1.55 | - | - |
| Retrain | 61.36±0.82 | - | - | - | 57.37±1.99 | - | - | - |
| FedEraser | 35.44±4.95 | **1.20±0.22** | 0.17±0.04 | 46.55s | 44.11±1.45 | 17.40±2.82 | 0.59±0.18 | 17.98s |
| FedRecovery | 53.41±1.14 | 7.64±1.02 | **0.05±0.01** | 14.93s | 48.73±3.11 | 26.16±2.10 | 0.60±0.14 | 14.93s |
| FedKDU | 61.09±0.27 | 3.90±0.40 | 0.06±0.02 | 43.54s | 66.59±2.98 | 17.22±1.69 | 0.14±0.06 | 17.11s |
| EWCSGA | 49.94±2.36 | 10.10±0.80 | 0.26±0.04 | 49.48s | 36.84±0.59 | 30.12±1.47 | 0.92±0.28 | 18.63s |
| MoDe | 54.34±0.41 | 5.21±0.52 | 0.06±0.01 | 44.64s | 58.64±3.31 | 18.55±1.82 | 0.50±0.15 | 16.61s |
| KNOT | 19.15±2.88 | 4.40±0.22 | 0.41±0.8 | **27.18s** | 22.38±3.49 | **4.15±0.42** | 0.99±0.20 | **10.16s** |
| FedSycle | **74.70±1.51** | 5.01±0.49 | **0.05±0.01** | 55.23s | **83.85±2.21** | 13.17±2.49 | **0.03±0.01** | 23.91s |

(b) Results on Par-2 data partition, running on VGG-16.

| Method | DomainNet | | | | PACS | | | |
|---|---|---|---|---|---|---|---|---|
|  | NT Acc. | ASR | DI | Efficiency | NT Acc. | ASR | DI | Efficiency |
| Pretrain | 73.23±1.02 | 99.71±0.44 | - | - | 91.84±0.15 | 99.89±0.04 | - | - |
| Retrain | 72.07±0.73 | - | - | - | 91.96±0.66 | - | - | - |
| FedEraser | 69.42±1.09 | **0.71±0.21** | 0.92±0.14 | 108s | 90.67±0.33 | **0.85±0.14** | 0.90±0.10 | 54.26s |
| FedRecovery | 73.74±1.40 | 64.95±2.76 | 1.44±0.37 | 106s | 73.74±3.35 | 46.20±1.84 | 1.08±0.22 | 49.98s |
| FedKDU | 69.11±0.64 | 14.72±1.18 | 0.35±0.14 | 100s | 89.69±0.92 | 1.16±0.14 | **0.11±0.02** | 44.24s |
| EWCSGA | 74.93±0.68 | 56.93±1.72 | 0.35±0.07 | 160s | 92.85±0.20 | 74.60±2.28 | 0.48±0.18 | 58.67s |
| MoDe | 74.71±0.55 | 21.98±1.94 | 0.40±0.09 | 103s | 92.63±0.15 | 16.49±1.19 | 1.28±0.22 | 46.16s |
| KNOT | 49.08±1.42 | 3.88±0.12 | 1.65±0.31 | **69.48s** | 62.02±4.88 | 2.45±0.41 | 1.13±0.25 | **29.91s** |
| FedSycle | **86.74±1.10** | 5.13±1.22 | **0.12±0.03** | 161s | **94.47±0.16** | 1.10±0.36 | 0.17±0.04 | 64.09s |

Table 6: Numerical Results of FU Algorithms. The best-performing algorithm is indicated in **bold**, and the second-best is shown in blue. (a) We designate the client possessing real (cartoon) domain data as the target client for DomainNet (PACS) dataset. (b) We designate the client possessing real and sketch (cartoon and photo) domain data as the target client for DomainNet (PACS) dataset.

Due to the non-competitive performance of the clustering-based method KNOT, we only report its related results in the appendix. Our analysis suggests that the KNOT method, which relies on performance competition between clusters and selects the aggregation result of one cluster as the global model, may encounter a mismatch in applicability in cross-silo scenarios where the number of clients is small and data heterogeneity is high, leading to subpar performance.

| Method | Pat-2 on MobileNet-V2 | | | Pat-2 on ResNet-18 | | |
|---|---|---|---|---|---|---|
| | NT Acc. | ASR | DI | NT Acc. | ASR | DI |
| Pretrain | 57.40±0.55 | 52.39±0.42 | - | 29.08±1.35 | 74.41±1.92 | - |
| Retrain | 75.94±1.88 | - | - | 30.84±1.67 | - | - |
| FedEraser | 68.16±1.12 | **4.31±0.09** | 0.44±0.07 | 32.57±0.33 | **7.44±1.20** | 0.25±0.04 |
| FedRecovery | 71.96±1.75 | 7.51±0.32 | 0.31±0.04 | 27.45±1.62 | 12.83±3.27 | 0.30±0.08 |
| FedKDU | 49.57±4.43 | 12.79±2.55 | 0.45±0.12 | 33.28±0.44 | 14.85±3.29 | 0.38±0.06 |
| EWCSGA | 40.51±2.34 | 19.08±1.14 | 0.24±0.06 | 25.55±1.53 | 15.31±1.07 | 0.51±0.19 |
| MoDe | 75.51±1.08 | 5.77±1.66 | 0.27±0.03 | 29.84±0.17 | 7.75±0.06 | 0.20±0.01 |
| FedSycle | **84.35±2.21** | 7.83±1.52 | **0.20±0.02** | **62.24±0.24** | 9.44±1.39 | **0.19±0.02** |

Table 7: Numerical Results of FU Algorithms on PACS. The best-performing algorithm is indicated in **bold**, and the second-best is shown in blue.

| Method | Dirichlet = 0.5 on MobileNet-V2 (PACS) | | |
|---|---|---|---|
| | NT Acc. | ASR | DI |
| Pretrain | 44.62±0.39 | 13.32±1.58 | - |
| Retrain | 86.92±2.78 | - | - |
| FedEraser | 82.39±2.25 | 0.81±0.29 | 0.24±0.07 |
| FedRecovery | 78.19±2.44 | 0.95±0.12 | 0.26±0.03 |
| FedKDU | 82.83±2.24 | 4.04±1.02 | 0.31±0.06 |
| EWCSGA | 80.82±1.25 | 3.35±0.82 | 0.42±0.13 |
| MoDe | 86.41±1.09 | **0.64±0.09** | 0.22±0.03 |
| FedSycle | **88.33±1.09** | 1.83±0.58 | **0.15±0.02** |

Table 8: Numerical Results of FU Algorithms. The best-performing algorithm is indicated in **bold**, and the second-best is shown in blue.

## B.8 CONVERGENCE EFFICIENCY ANALYSIS

To address inquiries regarding practical efficiency, we benchmarked the average wall-clock time for all baseline methods across all settings to reach their final converged state. The results, presented in Table 9, show that our method achieves a relatively **fast end-to-end training time cost** (2nd place of all), alongside better performance and fairness compared to the fastest method.

tabularx booktabs caption

Table 9: Comparison of end-to-end wall-clock time and performance. Our method provides a superior balance of efficiency and effectiveness.

| Method | Non-Target Accuracy (%) ↑ | Attack Success Rate (%) ↓ | Domain Inconsistency ↓ | End-to-End Time Cost ↓ |
|---|---|---|---|---|
| FedEraser | 59.91 | 5.04 | 0.65 | 2825s |
| FedRecovery | 62.41 | 36.24 | 0.79 | 2170s |
| FedKDU | 71.62 | 9.25 | 0.17 | 2558s |
| EWCGA | 63.64 | 42.94 | 0.50 | 2628s |
| MoDE | 70.08 | 15.56 | 0.56 | 3791s |
| **FedSyde (Ours)** | **84.94** | **6.10** | **0.09** | **2280s** |

## B.9 Generality and Scalability Experiments

To rigorously demonstrate that FedSyde can succeed in extreme FU scenarios and is broadly adaptable (W6), we conducted two supplementary experiments to illustrate its robustness and scalability.

### B.9.1 Generality of Domain Non-IID Setting

We conducted an experiment with 100 clients on DomainNet datasets (Dirichlet $\alpha = 0.5$) using MobileNetV2. The results in Table 10 confirm that FedSyde maintains its superior performance in a large-scale, domain-skewed environment.

Table 10: Performance on DomainNet with 100 Non-IID clients.

| Method | Non-Target Accuracy (%) ↑ | Attack Success Rate ↓ | Domain Inconsistency ↓ |
|---|---|---|---|
| FedKDU | 74.73±1.15 | 0.85±0.18 | 0.64±0.09 |
| MoDE | 73.98±0.77 | 1.33±0.39 | 0.81±0.14 |
| **FedSyde (Ours)** | **79.67±2.02** | **0.72±0.31** | **0.37±0.06** |

### B.9.2 Generality of General Visual Classification Setting

We conducted an experiment on CIFAR10 with 50 Non-IID clients, where each client has data from only 50% of the classes, using CNN with two convolutional layers. The results are shown in Table 11.

Table 11: Performance on CIFAR10 with 50 Non-IID clients.

| Method | Non-Target Accuracy (%) ↑ | Attack Success Rate ↓ | Domain Inconsistency ↓ |
|---|---|---|---|
| FedKDU | 69.18±2.12 | 4.10±0.35 | — |
| MoDE | 67.95±1.40 | 6.44±1.86 | — |
| **FedSyde (Ours)** | **86.78±0.93** | **5.19±0.64** | — |

## B.10 Quantitative Analysis of Synthesized Image Fidelity

We have conducted a quantitative analysis using standard image similarity metrics: Mean Squared Error (MSE), Peak Signal-to-Noise Ratio (PSNR), and Structural Similarity Index (SSIM). The setting is as follows:

- **Ours vs. Origin:** Average value between the generated images and the original same label images.
- **Same Label Image:** Average value between two different original images of the same label.
- **Gaussian Noise:** Average value before and after adding Gaussian noise with std=100.
- **Angel vs. AirCraft:** Use two different label data to compare the metrics.

As shown in Table 12, the generated images are quantitatively dissimilar from the originals, confirming they are not reconstructions. However, the competitive SSIM score demonstrates that abstract structure and style are preserved.

Table 12: Image Similarity Metrics. The generated image is quantitatively dissimilar to the original but retains high structural similarity.

| Type | MSE (↓ is more similar) | PSNR (dB) (↑ is more similar) | SSIM (↑ is more similar) |
|---|---|---|---|
| **Ours vs. Origin** | **7379** | **9.32** | **0.280** |
| Same Label Image | 7718 | 10.05 | 0.250 |
| Gaussian Noise | 6029 | 10.33 | 0.130 |
| Angel vs. AirCraft | 13357 | 7.35 | 0.120 |

- **Low Fidelity Proves No Reconstruction:** The very low PSNR (9.32 dB) and high MSE (7379) quantitatively prove that the generated image is not a faithful reconstruction of the original. In fact, it is **less similar to the original image than the same label image** from the original dataset on the MSE and PSNR metric.

- **High Structure Proves Utility:** The competitive SSIM (with Same Label Image) demonstrates that our method successfully preserves the abstract **structure and style** needed for effective retraining and Domain Inconsistency reduction.

## B.11 ABLATION STUDY ON THE ORIGINAL FL MODEL

To demonstrate that our performance gain is not solely from the pre-trained model (PTM), we empirically add a classification layer on top of the PTM without collaboratively training the original FL model. As shown in Table 13, we confirm that the large performance gain is primarily attributable to the powerful feature extraction capabilities of the pre-trained Q-Former. With the experiments, we believe the decision to retain the original, lightweight FL model was of good potential because it allows for a **fine-grained adaptation to the classification task** that a frozen PTM alone may not achieve, especially when it reaches the representing boundary of the PTM, and the **computational burden of the original FL model is relatively low**, which means it is a worthwhile trade-off.

Table 13: Ablation study on the contribution of the original FL model.

| Method | Non-Target Acc. (%) ↑ | Attack Success Rate ↓ | Domain Inconsistency ↓ |
|---|---|---|---|
| **FedSyde** | **84.94** | 6.10 | **0.09** |
| FedSyde w/o Origin FL Model | 83.15 | **5.98** | 0.14 |

## B.12 ANALYSIS OF FAIRNESS IN CLIENT CONTRIBUTIONS

We clarify that reducing domain inconsistency is precisely the challenge and goal that we focus on. This can lead to fairness in client contributions. Reducing domain inconsistency is crucial for the Federated Unlearning fairness and sustainability of FL systems. Table 14 shows a performance breakdown. While there is a 9% and 2% trade-off compared to the client-side only FedSyde in Non-Target Accuracy and Attack Success Rate, our full method still achieves the highest Non-Target Accuracy and top-2 Attack Success Rate. Crucially, FedSyde reduces domain inconsistency by 58.3%, which means much better fairness is achieved.

Table 14: Performance analysis on fairness and domain inconsistency.

| Method | Non-Target Acc. (%) ↑ | Attack Success Rate ↓ | Domain Inconsistency ↓ |
|---|---|---|---|
| Baselines Avg. | 50.84 | 5.60 | 0.12 |
| + client-side | 75.52 | 4.91 | 0.08 |
| **FedSyde** | **74.70** | 5.01 | **0.05** |

## B.13 ADAPTABILITY TO VIDEO ACTION RECOGNITION

To further demonstrate the adaptability of our method and address concerns, we extend the task to the video action recognition scenario. Specifically:

- Due to time constraints, we conduct the experiment on UCF-50, consisting of 50 action categories and over 6,000 videos.
- The data is split across 5 clients without label overlap. Training uses 16-frame random clip sampling. AdamW with learning rate=1e-4 and weight decay=1e-4, communication rounds=50, batch size=16, the client fraction=1, unlearning communication rounds=15, performance recovery rounds=10, others are the same as the main paper settings.
- We randomly select 16 frames to perform style-content decoupling using FedSyde, generate and train the synthesized frames accordingly.
- The ResNet50+LSTM neural networks are used to fit the task.
- We compute the Domain Inconsistency metric according to Equation 3 of the main paper. Since there is no domain categories in UCF-50, the domain is defined as the label in this experiment.

As can be seen in Table 15, **FedSyde demonstrates superior performance across all metrics**, highlighting the potential of it for broader applications in vision-based tasks.

Table 15: Performance on Video Action Recognition (UCF-50).

| Method | Non-Target Accuracy | Attack Success Rate | Domain Inconsistency |
|---|---|---|---|
| Pretrain (FedAvg) | 73.21±0.76 | 48.33±0.41 | - |
| Retrain (FedAvg) | 76.96±1.12 | - | - |
| FedKDU | 74.50±0.48 | 1.10±0.28 | 0.32±0.04 |
| **FedSyde** | **81.54±1.06** | **0.72±0.15** | **0.11±0.02** |

## C  DECLARATION OF LLM USAGE

The LLM is used only for writing, editing, or formatting purposes and does not impact the core methodology, scientific rigorousness, or originality of the research.

