# OpenReview forum: "FedSycle: Mitigating Post-Unlearning Performance Inconsistency in Federated Learning via Latent Feature Decoupling"
_ICLR.cc/2026/Conference — ICLR 2026 Conference Withdrawn Submission_

### Official Review · Reviewer_c2w1 · 2025-10-28

**Soundness:** 2
**Presentation:** 1
**Contribution:** 2
**Rating:** 4
**Confidence:** 4

**Summary:**

This work identifies a practical fairness issue in federated unlearning: removing one client's domain-specific data (e.g., sketches) can disproportionately harm the performance for remaining clients (e.g., real photos). To address this, the proposed "FedSycle" framework relies heavily on pre-trained models (Q-Former and Stable Diffusion) to perform a style-content decoupling. Only "style" attributes are centrally aggregated, allowing the server to synthesize a domain-balanced auxiliary dataset for retraining without seeing private content.

**Strengths:**

This manuscript describes the federated unlearning technique and state-of-the-art methods in considerable detail, and the ablation experiments are also relatively comprehensive and provide necessary theoretical validation.

**Weaknesses:**

1. Comparisons with more recent unlearning or heterogeneity-aware federated learning baselines (2024–2025) are missing, which slightly weakens the empirical strength of the paper. Including up-to-date methods would provide a more convincing demonstration of the proposed approach’s effectiveness.

2. It would be beneficial to further validate the proposed method on the Office-Caltech dataset across multiple domains to assess its generalization capability under diverse distributional conditions.

3. Several detail-related issues are present throughout the manuscript. For example, the citation at line 943 fails to render correctly; the notations used to represent datasets are inconsistent; and the internal references to Appendix A (line 361) and Appendix B are incorrect. These should be carefully revised for accuracy and consistency.

4. The manuscript states that “FedSycle operates by decoupling client data into distinct latent representations: one capturing semantic content (retained locally for privacy and to boost client-side retraining efficiency) and another capturing domain-specific attributes (e.g., texture, color).” It would significantly strengthen the paper to include corresponding visualization results (e.g., latent feature maps or t-SNE plots) to empirically validate this decoupling effect.

5. The visualization of generated images in Figure 4 raises some concerns. The paper states that the server receives only the uploaded “style” representations—mainly texture and color—yet the generated images appear to contain semantic content (e.g., clear object structures such as houses in the first image). Please clarify why the generated outputs include such semantic details if only style information is transmitted.

6. Regarding the Par-2 data partition setting, it seems feasible to extend the experimental setup to include more than four clients. Conducting experiments with a larger number of clients could better demonstrate the scalability and robustness of the proposed framework.

**Questions:**

Please refer to above

---

### Official Review · Reviewer_Nn2S · 2025-10-30

**Soundness:** 3
**Presentation:** 3
**Contribution:** 3
**Rating:** 6
**Confidence:** 3

**Summary:**

The paper proposes FedSycle, a novel federated unlearning (FU) framework designed to mitigate post-unlearning performance inconsistency—a newly identified fairness issue in which the unlearning of one client causes uneven accuracy shifts across the remaining clients’ domains. FedSycle introduces three key ideas: 1) Domain inconsistency (DI) metric, which is a quantitative measure of domain-level performance variance after unlearning; 2) Style-content decoupling – using a pretrained Q-former to separate private content features (kept locally) from transferable style features (shared for aggregation); 3) Server-side auxiliary generation – synthesizing domain-aligned auxiliary data via diffusion models to rebalance domain performance. The paper provides convergence analysis and extensive experiments on PACS and DomainNet, showing large reductions in DI (up to 83%) and substantial accuracy improvements over baselines such as FedEraser, FedKDU, and MoDe.

**Strengths:**

+ The fairness-oriented motivation (ensuring consistent post-unlearning performance across clients) is timely and practically relevant.
+ Integration of pretrained models (Q-Former + stable diffusion) for lightweight recovery is well-reasoned and technically sound.
+ The paper provides a convergence guarantee under standard smoothness and bounded-variance assumptions. The approach maintains privacy by transmitting only non-invertible style features.

**Weaknesses:**

+ Using Q-Former and diffusion generators may limit applicability in resource-constrained FL settings. Some lightweight or alternative decoupling strategies should be discussed. While several FU algorithms are compared, other recent works (e.g., feature-space unlearning, asynchronous unlearning) are not included.

- The paper assumes style features are privacy-safe, but does not empirically evaluate reconstruction risks or communication cost.

**Questions:**

- Could FedSycle be adapted to class-level or sample-level unlearning, and would the DI metric remain valid?
- How computationally demanding are Q-Former and diffusion models during training? Any estimates of FLOPs or communication overhead?
- Have you quantified privacy leakage from uploaded style centroids (e.g., via inversion attacks)?
- Can the server-side auxiliary generation be replaced with shared synthetic prototypes to further reduce cost?

---

### Official Review · Reviewer_n8qc · 2025-10-31

**Soundness:** 1
**Presentation:** 2
**Contribution:** 3
**Rating:** 2
**Confidence:** 3

**Summary:**

The paper proposes a framework for client-level Federated Unlearning under domain shift termed FedSycle. FedSycle uses a pretrained QFormer to separate client data into a content matrix $I_c$ and a style matrix $I_s$. Content is kept local and used for post target removal retraining, while style gets clustered and the centroids of the clusters are uploaded to the server. The server combines the uploaded style centroids with label prompts and uses a pre-trained Stable Diffusion model to generate auxiliary images for every centroid–label pair (M×L synthetic images per client). A server step is run on this synthetic data and that update is mixed with the aggregated client updates using a tunable weight $\alpha$. This aims to “recalibrate” performance across domains so post-unlearning shifts are more uniform. The paper also introduces a variance based metric, “Domain Inconsistency”, to quantify uneven post-unlearning performance shifts across domains. Experiments are conducted on DomainNet and PACS across various architectures/settings/heterogeneity set-ups.

**Strengths:**

1. Clear foundational motivation and problem significance. Effective FU is an important milestone for real-world FL deployment.

2. Method intuition and design: Using a pretrained QFormer model to decouple content (kept local) and style (shared as clustered centroids) is a neat idea for minimizing privacy exposure while enabling server-side diffusion-guide synthetic replay tailored to domain shift. The dual-side design is coherent and aligns well with the stated objective.

3. Results presented show a healthy improvement over the baselines (with some caveats as detailed below).

**Weaknesses:**

1. **Flawed specific motivation**. While, as stated in S1, the general problem of FU is significant, the authors attack a subset of it which seems more like an engineered problem and less a problem encountered in real life: If the goal is unlearning at the client-level, I would be biased towards seeing a solution tailored to cross-device rather than cross-silo FL. The paper frames Domain Inconsistency (uneven post-unlearning performance across domains) and evaluates primarily in a cross-silo-like regime where each client is aligned to one or two domains. While a reasonable first step, it would help to motivate real cross-silo use cases where client-level RTBF triggers unlearning, and to complement the main results with more cross-device-like settings in the main paper. The choice “same number of clients as domains” narrows generality and may accentuate the phenomenon the method is designed to fix.

2. **Effect of M** The method clusters per-client style features into M centroids (Eq. 4), and the server uses style centroids with label prompts to synthesize M×L auxiliary images. However, I could not find how M is selected or whether a sensitivity sweep exists. Please (i) report the values of M per dataset/backbone, (ii) add M to Algorithm 1’s inputs, and (iii) provide a sensitivity analysis or a simple selection heuristic since M directly affects both compute (generation) and performance. Figure 2 Represents each client as owning images originating from a series of very distinct domains, in which case it would be trivial to find the optimal $M$, however in practice this would be difficult and I imagine quite important for downstream performance.

3. **Experimental evaluation 1**
Section 5.1 states a randomly selected target client “without loss of generality,” but Table 1’s captions specify particular target domains (“real” for DomainNet, “cartoon” for PACS) rather than random. Given DI is defined precisely to capture domain-dependent shifts, results will vary with which domain is removed. Please report per-removed-domain results (and their average/variance), or at least provide aggregated results over several target clients to support the “without loss of generality” claim. Even the motivation in Section 1 is built upon unlearning of different domains affecting the result disproportionately. This is a very significant omission that greatly undermines the paper.

4. **Experimental evaluation 2**
Retraining from scratch without the target clients should intuitively provide an upper bound for the algorithm, if all hyper-parameters are configured correctly (see also FedEraser page 5 in support of this). Can the authors explain why/how in the presented experiments they outperform retraining by up to 40% accuracy?

5. **Experimental evaluation 3**
The authors state "based on the cross-silo scenario, without loss of generality, we select the same number of clients as the dataset domains". To support the generality claim, the authors include some results (albeit in the appendix) with the Dirichlet partioning strategy and more clients (50-100). This is relegated to the appendix, Tables 11 and 12, but I fail to understand why only a subset of the baselines are presented alongside the proposed method. I believe a more in-depth investigation into performance as the number of clients varies is necessary.

6. **With regards to DI**, it would help to clarify numerical stability when the denominator is small, and to discuss whether any measure are necessary/put in place to avoid giving any domain disproportionate influence (e.g. due to size).

Typos
- on line 291 (Q-Former DeaDiff te pqi2024deadiff)
- line 421 Lower the ASR, better the unlearn effectiveness. Same for the following bullet points, please improve the grammar.
- line 943 As shown in Tab. ??,
- line 1018 the proposed method's number is bold while it is outperformed by FedEraser in ASR.

**Questions:**

1. It would help if the authors could provide concrete cross-silo use cases for client-level unlearning beyond individual RTBF. This would strengthen the motivation for a cross-silo focus as RTBF only applies to individuals, not companies.
2. Can the authors provide some references and arguments for the claim "the extraction and transmission of class prototypes inevitably exposes sensitive client data, including labels and discriminative image features"? This is quite a disseminated approach in multiple areas of FL, so the claim, which forms the foundation for the paper, needs substantiating. An empirical audit supporting this leakage risk, and clarification on how style centroids mitigate it would reinforce why prototype sharing is less suitable than style sharing, thus strengthening the rationale.
3. Convergence analysis: The theorem provides a non-convex convergence rate under smoothness/variance/dissimilarity assumptions, plus an assumption bounding the gap between server and true gradients. Could you clarify how this result informs practical hyper-parameter choices, and whether the bound meaningfully connects to DI reduction.

---

### Official Review · Reviewer_kYvy · 2025-10-31

**Soundness:** 2
**Presentation:** 2
**Contribution:** 2
**Rating:** 4
**Confidence:** 3

**Summary:**

The paper proposes a method for federated unlearning in which the training data spans multiple domains, and a client can request the removal of its data. The paper aims to ensure that the accuracy of the unlearned model across clients is similar to ensure fairness. The algorithm depends on decoupling the style and the content.

**Strengths:**

1. Federated unlearning is an interesting and important problem.
2. Overall, the paper is well written and easy to follow.

**Weaknesses:**

My primary concern is that if the method actually unlearns.
* The main results in Table 1 (and more results in Appendix Tables 6, 7, 8) on DomainNet show that the unlearning is weak compared to other baselines. Though for PACS, the method seems to perform well, PACS has a class imbalance, so not sure how reliable the metrics are. It needs more elaboration.
* In Table 2, it seems the server-side training reduces the unlearning capability; compare ASR for the first and third row. Though DI improves in the third row, is it at the expense of weaker unlearning?

More:
1. Please use consistent symbols: F is used as a loss function in eq 1 and as the feature extractor in Fig 2.
2. Improve the definition of DI in Definition 1 (Line 238) with a better definition of $D_r$ and $D_\phi$. Does the metric assume that a domain is fully contained in a single client? DI is only used in experiments; it feels it better fits in the experiments.
3. I think it is expected that unlearning will have a different effect on the marginal accuracy of different domains and different classes. It is possible that a class is easier to separate when a particular domain is absent, and thus its marginal accuracy improves after unlearning. Is there any particular application where the fairness is more important than the accuracy improvement in certain domains/classes?
4. Can the authors explain style-content decoupling in more detail, including some more information on Q-former? The paper refers to Deadiff without explaining what it is. The main technical content in 4.1 and 4.2 depends on these details.
5. Can the authors add more details to Fig 3? For example, if the black arrows denote client gradients, then the average may not be the blue arrow. Also, how to define the domain consistency area, does it have any relation with DI metric? How do we interpret it? How to define this for an application?
6. From the description in Section 4.3, I'm confused about how the server-side training reduces domain inconsistency. For example, it does not use any loss component to reduce inconsistency.
7. The paper claims to have results on ResNet-18, whereas the main document does not have any results on that.

**Questions:**

As above.

---

### Note · Authors · 2025-11-14

I have read and agree with the venue's withdrawal policy on behalf of myself and my co-authors.